# HKT1;1 and HKT1;2 Na^+^ Transporters from *Solanum galapagense* Play Different Roles in the Plant Na^+^ Distribution under Salinity

**DOI:** 10.3390/ijms23095130

**Published:** 2022-05-04

**Authors:** Maria J. Asins, Maria R. Romero-Aranda, Jesus Espinosa, Paloma González-Fernández, Emilio Jaime-Fernández, Jose A. Traverso, Emilio A. Carbonell, Andres Belver

**Affiliations:** 1Instituto Valenciano de Investigaciones Agrarias (IVIA), 46113 Moncada, Spain; carbonell.emi@gmail.com; 2Department of Plant Breeding and Biotechnology, La Mayora Institute for Mediterranean and Subtropical Horticulture (IHSM), UMA/CSIC, 29750 Algarrobo-Costa, Spain; rromero@eelm.csic.es (M.R.R.-A.); emilio@ihsm.uma-csic.es (E.J.-F.); 3Department of Stress, Development and Signaling of Plants, Estación Experimental del Zaidín (EEZ), Consejo Superior de Investigaciones Científicas (CSIC), C/Prof. Albareda 1, 18008 Granada, Spain; jesus.espinosa@eez.csic.es (J.E.); paloma.gonzalez.fernandez@hotmail.es (P.G.-F.); andres.belver@eez.csic.es (A.B.); 4Department of Cellular Biology, Faculty of Sciences, University of Granada, 18071 Granada, Spain; traverso@ugr.es

**Keywords:** tomato, *S. cheesmaniae*, salt tolerance, rootstock breeding, yield, fruit quality, K^+^ and Na^+^ homeostasis, long-distance transport

## Abstract

Salt tolerance is a target trait in plant science and tomato breeding programs. Wild tomato accessions have been often explored for this purpose. Since shoot Na^+^/K^+^ is a key component of salt tolerance, RNAi-mediated knockdown isogenic lines obtained for *Solanum galapagense* alleles encoding both class I Na^+^ transporters HKT1;1 and HKT1;2 were used to investigate the silencing effects on the Na and K contents of the xylem sap, and source and sink organs of the scion, and their contribution to salt tolerance in all 16 rootstock/scion combinations of non-silenced and silenced lines, under two salinity treatments. The results show that SgHKT1;1 is operating differently from SgHKT1;2 regarding Na circulation in the tomato vascular system under salinity. A model was built to show that using silenced *SgHKT1;1* line as rootstock would improve salt tolerance and fruit quality of varieties carrying the wild type *SgHKT1;2* allele. Moreover, this increasing effect on both yield and fruit soluble solids content of silencing *SgHKT1;1* could explain that a low expressing *HKT1;1* variant was fixed in *S. lycopersicum* during domestication, and the paradox of increasing agronomic salt tolerance through silencing the *HKT1;1* allele from *S. galapagense*, a salt adapted species.

## 1. Introduction

Salinity in soils and in water available for irrigation is one of the main environmental factors limiting the growth and yield of crops, including tomato. The estimated cost of losses in agricultural production due to salinity worldwide is 27 billion dollars per year [1], leading in some regions to increased poverty and food import dependence. Tomato is the most important horticultural crop in terms of yield and cultivated land but exerts intense pressure on water resources because it requires large amounts of water to grow [2], explaining the growing need to use low-quality irrigation water. Therefore, salt tolerance is a target trait in both plant science and tomato breeding programs.

Wild tomato species have been often explored to search for salt-tolerant accessions [3,4,5,6]. *S. cheesmaniae,* and *S. galapagense*, previously considered to belong to the same species of *Lycopersicon cheesmanii* by [7], are endemic to the Galapagos Islands and adapted to highly saline coastal habitats, showing some halophytic features such as a lack of toxicity in Na^+^ leaf accumulation [8,9]. Other interesting features of these Galapagos wild species regarding tomato breeding are their high sugar and β-Carotene contents, the jointless pedicel, and the tolerance/resistance to certain pathogens [10]. In addition, comparing tolerant accessions from *S. pimpinellifolium* and *S. cheesmaniae*, only those from the latter species could yield fruits under the highest salinity level tested (325 mM NaCl, 30 dS/m electric conductivity) [4]. Recently, Ref. [5] studied the variability for 20 traits reflecting growth, physiology, and ion content of 67 accessions from *S cheesmaniae* and *S. galapagense* and found three accessions showing high salt tolerance in terms of growth, two of them from *S. galapagense*. To efficiently exploit the wild germplasm in tomato breeding programs, knowledge of the genomic regions containing the beneficious alleles to improve salt tolerance is needed, and this information, among others, is obtained by quantitative trait loci (QTL) analysis in segregant populations [11]. With this mining purpose, salt-tolerant accessions from *S. pimpinellifolium*, *S. cheesmaniae*, and *S. pennellii* have been used as parents of segregating populations where the inheritance of salt tolerance has been studied to determine the (wild vs. cultivated) origin of beneficial alleles or haplotypes [12,13,14,15,16,17]. However, salt tolerance is a complex trait, presenting numerous physiological components and variations during plant development and with the level of salinity. One important component of salt tolerance in plants, including tomato, is the maintenance of cellular Na^+^/K^+^ homeostasis, essentially through Na^+^ exclusion from the root, modulation of root-shoot transport of Na^+^ and K^+^, and cellular compartmentalization of Na^+^ [18,19].

Two tomato genes, encoding class I HKT transporters HKT1;1 and HKT1;2 [17], which were shown to be Na^+^ selective transporters [17,20], were proposed to underlie a major locus (QTL) on chromosome 7, controlling shoot Na^+^/K^+^ in two RIL populations derived from *S. lycopersicum* × *S. pimpinellifolium* and *S. lycopersicum* × *S. cheesmaniae* [13,17,21,22]. Given the tight linkage between HKT1;1 and HKT1;2 loci in tomato [17], a reverse genetic strategy based on loss of gene function was necessary to determine which HKT1 transporter, if any, plays the main role in regulating Na^+^/K^+^ shoot concentration when cultivated under saline conditions. Using two near-isogenic lines (NILs) that were homozygous for either the *S. lycopersicum* allele (NIL17) or for the *S. cheesmaniae* allele (NIL14) at both linked HKT1 loci and the transgenic lines derived from these NILs in which each *HKT1;1* and *HKT1;2* genes were silenced by stable transformation, Ref. [23] showed that RNAi-mediated knockdown any of *S. lycopersicum* or *S. cheesmaniae* alleles at HKT1;2 altered the leaf Na^+^/K^+^ ratio and caused hypersensitivity to salinity in plants cultivated under transpiring conditions, whereas silencing these alleles at HKT1;1 had a lesser effect. Moreover, using the silenced *HKT1;2* transgenic lines derived from both NILs, Ref. [24] showed that the Na^+^ transporter protects the flower from Na^+^ toxicity and mitigates the reduction in tomato fruit yield under salinity conditions. However, Ref. [23] noted a significant growth increase in the NIL14 lines silenced for *ScHKT1,1*, with respect to the non-silenced genotype, under controlled conditions in a culture chamber, reaching a growth level similar to that of NIL17 in the absence of NaCl. The fact that both the cheesmaniae and pimpinellifolium alleles at HKT1;1 are hyperactive in leaf, contrary to that of lycopersicum, seems to indicate that the cultivated tomato species could have diverged from these wild species not only fixing a hyperactive allele at HKT1;2 in the root, but also decreasing the expression of HKT1;1 in leaf. In general, loss-of-function HKT mutants are often associated with salt-sensitive phenotypes [25], but exceptions have been reported. In maize, a domestication-associated reduction in a K^+^-preferring HKT transporter activity underlies maize shoot K^+^ accumulation and salt tolerance [26]. Additionally, RNAi-mediated down regulation of HvHKT1;5 from barley resulted in salt tolerance [27]. Therefore, it is important to investigate the possibly different roles of *S. cheesmaniae* HKT1;1 and HKT1;2 in mature leaves, the main source organs where xylem-to-phloem solute transfer happens, and roots, where solute transfer in the opposite direction from the phloem to the xylem mostly occurs [28]. In a previous study using data from 10 selected rootstock/scion combinations out of a total of 16 derived from combining four isogenic lines (non-silenced, single RNAi-silenced lines for *ScHKT1;1* and *ScHKT1;2*, as well as a silenced line at both loci from the near-isogenic line NIL14), Ref. [29] concluded that, (1) in addition to xylem Na^+^ unloading, ScHKT1;2 could also be involved in Na^+^ uploading into the phloem thus promoting Na^+^ recirculation from aerial parts to the roots, and (2) ScHKT1;1 function loss in the rootstock minimized yield loss and improved some fruit quality parameters under salinity. The present study, using the same data but from all 16 rootstock/scion combinations and an approach that investigates the interactions between all the factors involved, further supports those conclusions and provides a model for Na^+^ recirculation that could assist the design of rootstock/scion combinations in improving salt tolerance and the fruit contents of soluble solids and β-carotenoids in tomato.

## 2. Results

The salt tolerant accession L2 of *S. cheesmaniae* from which NIL14 was obtained, has been now re-classified as *S. galapagense* because of its leaf morphology (Appendix A).

### 2.1. Effects of Salinity Treatment on Na and K Contents of the Different Tissues and Their Relationships

Salinity increased the Na concentration in the xylem sap, leaves, flower inflorescence, and fruits (X_Na, L_Na, Fl_Na, and F_Na, respectively) of all rootstock/scion combinations (Appendix A). This increment was particularly small in fruits of the h1/Wt (silenced *SgHKT1;1* rootstock/Wildtype scion) combination. Salinity generally reduced K in leaves (although this reduction was quite small in h1/h1) and induced different changes in X_K and F_K depending on the rootstock/scion combination, particularly in fruit where the K content increased under salinity (Wt/h1-h2, h1-h2/h1-h2, h2/h1-h2, h2/Wt and h1/h1-h2) (Appendix A). Salinity generally increased the Na/K ratio in all tissues sampled, being that this increment was remarkably small in h1/h1 leaves and h1/Wt fruits (Appendix A). Under the control conditions, no mean differences were found for all 16 combinations in the Na tissular contents and the Na/K ratio of leaf, fruit, and xylem sap. Therefore, salinity mostly increased Na and Na/K ratio in xylem sap, leaf, and fruit and changed (increased or decreased) K in fruit depending on the rootstock/scion combination of silenced genes.

Regarding Na and K distributions among tissues, Figure 1A shows a schematic representation of significant correlations connecting Na and K contents of leaves (L), flowers (Fl), fruits (F), and xylem sap (X). Under salinity, the highest positive correlation for Na content was found between L and Fl, while for K, it was between X and F (0.78 and 0.87, respectively, in Appendix A and blue lines in Figure 1). The correlation between the content of K in flower and the [K^+^] in the xylem sap is similar in the two salinity treatments suggesting that the xylem is a usual via for K flower nutrition. Leaf K content is not related to the K content of other tissues. Strong negative correlations between L_K and Fl_Na (r = −0.87), X_Na (r = −0.72) and L_Na (r = −0.70) were found under salinity.

As expected, the contents of Na in leaf, fruit and xylem sap were highly correlated (r = 0.97, 0.89 and 0.72, respectively, in Appendix A and black lines in Figure 1A) between salinity levels (0 and 70 NaCl mM), suggesting that salinity mainly changed K tissular distributions accordingly to Na increments trying to maintain Na/K ratios (Na/K homeostasis) (Figure 1B).

### 2.2. Rootstock and Scion Contributions of S. galapagense Alleles at HKT1;1 and HKT1;2 to Na Distribution, Salt Tolerance and Fruit Quality

Means and standard errors of all rootstock/scion genotypic combinations for vegetative and agronomic (fruit-related) traits for both treatments are in Appendix A. Salinity reduced aerial plant growth (measured as the dried weight of the aerial part of the plant, APDW) and total fruit yield (TFW). Three remarkable facts were observed: (1) the fruit yield of the h2/h2 combination was particularly low under control conditions, (2) the salinity-related reduction in the number of fruits (TFN) was not significant in the h1/h1 and h1/Wt combinations, and (3) the salinity increased the RDW of h1/Wt. Taking these results together suggests that, in contrast to the biological effects of silencing *SgHKT1;2*, silencing *SgHKT1;1* does not have a negative effect on NIL14.

To study the relationships among the traits, correlations and principal component analyses were performed. There was no significant correlation between the control and salinity (in yellow in Appendix A) for traits related to fruit yield (TFW, TFN) or vegetative growth (APDW, RDW). Interestingly, salt tolerance (evaluated as total fruit yield under salinity) had a highly negative relationship to L_Na/K (r = −0.80), Fl_Na (−0.77), and L_Na (−0.76), and was positively related to L_K (0.68) (Appendix A and Figure 2B).

Considering the 16 rootstock/scion combinations of silenced and wild type *S. galapagense* HKT1 lines, SSC was highly and positively correlated to the fruit yield (TFW and TFN) and vegetative growth (APDW, RDW) only under salinity (Appendix A), which explains their grouping together in the principal component analysis (Figure 2B). Therefore, a better vegetative growth was related not only to a greater fruit yield but also to a higher SSC of the fruits in these materials under salinity.

Given that the four lines (Wt, h1, h2, and h1-h2) only differ in the silencing (or not) of the *S. galapagense* allele at loci HKT1;1 and HKT1;2, it is possible to test their main effects and interactions at the rootstock (R1 and R2), at the scion (S1 and S2), and salinity (E) by the statistical analysis. Many effects and interactions were significant, particularly for L_Na, (Appendix A), denoting the complexity of the physiology behind this trait. Rootstock genotype at HKT1.1 (R1) and HKT1.2 (R2) were involved in three interactions for the total fruit yield.

(TFW): R2 × E, R2 × S1, and R1 × S1 (Figure 3A,B,C, respectively). Arrows in all figures show the direction of the effects of silencing *SgHKT1;1* at the scion (orange arrows) and at the root (blue arrows), and silencing *SgHKT1;2* at the scion (green arrows) and at the root (black arrows). The significant interaction of R2×E for TFW (Figure 3A) corresponds to the fact that the direction of silencing effects depends on the treatment (i.e., upwards under control, downwards under salinity). The significant interaction between the rootstock HKT1.2 and scion HKT1.1 genotypes (R2 × S1) for TFW is visualized in Figure 3B, where silencing HKT1;1 at the scion (orange arrow from S1-Wt to S1-h1) improves the fruit yield of the silenced HKT1;2 genotype at the rootstock (R2-h2) but has no change in the wild type genotype (R2-Wt), and it happens regardless of the treatment since the triple R2 × S1 × E interaction was not significant. Most importantly, the best, high-yielding plant combined the silenced genotype at HKT1;1 (h1) at both the rootstock (R1) and the scion (S1), as shown in the significant R1 × S1 interaction for TFW (Figure 3C), regardless of the treatment since the triple interaction R1 × S1 × E was not significant. This h1:h1 combination (R1×S1 interaction) also showed the least Na leaf content (Figure 3D) and the highest β-Carotenoid fruit concentration (Figure 3E). It is worthy to note that the R1 × S1 interaction was not significant for SSC (Appendix A).

Triple interactions involving the salinity level for Na content in the different tissues were studied, and significant results are shown in Figure 4. In all cases, no mean differences were found for control conditions (red squares), probably because of the low amounts (magnitude or scale effect). The triple interaction of R1 × S1 × E for L_Na was significant (Appendix A), but the effects of silencing under salinity are similar to those described for the R1 × S1 interaction in Figure 3D. The directions of the silencing effects at HKT1;1 (orange arrows) and HKT1;2 (green arrows) at the scion on L_Na are opposite (Figure 4A), and these directions change when considering the effects on F_Na (Figure 4B). Similarly, the directions of the silencing effects at HKT1;1 (blue arrows) and HKT1;2 (black arrows) at the rootstock on L_Na are also opposite (Figure 4H). Thus, against expectations, silencing *SgHKT1;1* decreases the leaf Na content (Figure 3D, Figure 4H, and Appendix A).

Regarding the silenced galapagense allele at locus HKT1;2 in the scion, salinity increased the Na content in flowers and leaves but reduced it in fruits (S2 × E interaction) (Appendix A). Simultaneously, TFW, APDW, RDW, and fruit SSC were reduced while fruit juice acidity increased. On the other hand, there was no noticeable difference under the control conditions.

Salinity increased fruit SSC in most rootstock/scion combinations (as it was shown in Appendix A). However, considering individual effects, we observe that under salinity, silencing *SgHKT1;2* at the scion (S2 × E interaction) (green arrows in Appendix A) or at the rootstock (R2 × E) (black arrows in Appendix A) both reduced SSC. On the other hand, silencing *SgHKT1;1* at the scion (S1 × E interaction) (orange arrows in Appendix A) or at the rootstock (R1 × E interaction) (blue arrows in Appendix A) both increased SSC under salinity and decreased it under control (Appendix A). The significant interaction of S1 × S2 for SSC showed that the effect of silencing *SgHKT1;1* on SSC depended on the scion genotype at HKT1;2 (orange arrows in Appendix A): if *SgHKT1;2* is silenced (in S2-h2), SSC decreases (when changing from S1-Wt in blue to S1-h1 in red), while if it is the S2-Wt, SSC increases. Therefore, the wild type allele *SgHKT1;2* at the scion (S2-Wt in Appendix A) and at the rootstock R2-Wt (Appendix A) are apparently relevant for the fruit soluble solids content in NIL14. As a result, the best SSC and Beta-carotenoid increasing scion combined wild *SgHKT1;2* and silenced *SgHKT1;1* (Appendix A), and the silenced *SgHKT1;1* genotype at the rootstock (for β-carotenoid in Figure 3E).

### 2.3. Silencing Rootstock and Scion S. galapagense Alleles at HKT1;1 and HKT1;2 Affect K^+^ Tissular Distribution under Both Control and Salinity Conditions in a Complex Way

All correlations involving the K leaf content under salinity in Figure 1 disappear under the control conditions where the leaf K contents of all rootstock/scion combinations have higher means when compared to their values in salinity (Appendix A). Under an external Na^+^ excess, our results show that the final leaf K content depends on several factors, directly or indirectly, related to the genotype at both HKT1 loci in root and scion.

Silencing the *S. galapagense* alleles at HKT1;1 and HKT1;2 loci clearly affected the leaf K content (L_K in Appendix A). In addition, several significant and interesting double and triple interactions were observed (Appendix A) for X_K (S1 × S2, S1 × S2 × E), L_K (R1 × R2, R1 × S1, S1 × S2, R1 × R2 × E, R1 × S1 × E), Fl_K (S1 × S2), and F_K (S1 × S2, S1 × S2 × E). Silencing *SgHKT1;2* at the rootstock decreased L_K independently of the genotype at HKT1;1 (black arrows in R1 × R2 and R1 × R2 × E interactions in Appendix A), even under the control conditions (and wild type *SgHKT1;1*). This result suggests that SgHKT1;2 at the root is somehow related to the xylem K load under the control and, particularly, under salinity. Conversely, silencing *HKT1;1* at the root (blue arrows in R2 × R1, R1 × R2 × E, S1 × R1, and R1 × S1 × E interactions) increases L_K, suggesting HKT1;1 at the root is associated with xylem K unload. Simultaneous to the L_K increment, silencing *SgHKT1;1* at the root increases RDW (if R2 is Wt) and TFW (if it is also silenced at the scion).

In the scion, silencing *SgHKT1;2* (green arrows) increases X_K (even under control conditions), Fl_K, and F_K (when S1 is h1, mostly) but also decreases L_K when S1 is h1. The effect of silencing *SgHKT1;1* (orange arrows) depends on the genotype at HKT1;2 at the scion (S2), and at HKT1;1 at the root (R1). Thus, if HKT1;2 is Wt, silencing *SgHKT1;1* decreases [K+] in the xylem sap and fruit K content (no difference was observed for both the leaf and flower), while if HKT1;2 is silenced too, then [K^+^] increases in the xylem, flower, and fruit, but decreases in leaves (S1 × S2 and S1 × S2 × E interactions for X_K, L_K, Fl_K, and F_K in Appendix A). However, if *SgHKT1;1* is also silenced in the root (R1 × S1interaction for L_K), silencing *SgHKT1;1* at the scion increased L_K. Therefore, the tissular distribution of K is directly or indirectly regulated by genotypic interactions between SgHKT1;1 and SgHKT1;2.

## 3. Discussion

In order to interpret the effects of silencing *SgHKT1;1* and *SgHKT1;2* in terms of their putative roles at the whole plant level, we need to build up a model (Figure 5) that allows the integration of the results of this reciprocal grafting experiment, the previous knowledge on those transporters, and the main features of plant vasculature. Following [28], plants contain two parallel aqueous pathways for intercellular solute movement, separated by the plasma membrane. The aqueous phase of the apoplast, outside the plasma membrane and consisting almost entirely of cell walls and the xylem (red in Figure 5), and the symplasm, which lies within the plasma membrane, is connected from cell to cell by the plasmodesmata and consists of living nucleated cells and of the enucleate conducting cells of the phloem (blue in Figure 5). Flow in the xylem is essentially unidirectional from roots to sites of water evaporation (leaves manly), while movement in the phloem proceeds from sites of assimilate production, primarily mature leaves, to sites of utilization in expanding tissues (new leaves, flowers, and roots) and storage (fruits) sinks. The characteristic side-by-side arrangement of the xylem and the phloem through the plant facilitates solute exchange between the two, especially in the highly branched vascular system of source (leaf) and sink (root) regions. Thus, solute transfer between the xylem and the phloem would play an important role in nutrient (K, Na) partitioning between organs. The minor vein endings in transpiring leaves are an important site for xylem-to-phloem solute transfer. Thus, the nutritional content of the sieve tube sap is enriched at the expense of the transpiration stream (circle at leaf in Figure 5). Solute transfer in the opposite direction from the phloem to the xylem occurs largely in the root (circle at root in Figure 5). Thus, a substantial proportion of some solutes (i.e., Na, K) delivered to the root by the phloem may be recirculated back to the shoot via xylem. Therefore, it could be reasonable to simplify by saying that the nutrient content of the fruit basically depends on what arrives via phloem from mature leaves, while the leaf nutrient content depends on what is found from the xylem and what exports through the phloem (and may come back from the root via xylem again), at least under the control conditions. Under salinity only, the fruit K content was highly correlated to the xylem sap [K^+^] (0.87, Appendix A and Figure 1A). Since a xylem backflow from the fruit to the plant has been observed during truss development, that could represent a net loss under water stress [30], a concomitant K^+^ efflux from the fruit (the major K plant reservoir) to the xylem sap could be hypothesized to explain the correlation between F_K and X_K under salinity (discontinuous red line in Figure 5).

Tomato *HKT1;1* and *HKT1;2* genes encode class I HKT Na^+^-selective transporters [17,20], which are expressed in the vascular system, in the xylem and possibly in the phloem of tomato root and shoot tissues [23]. Sequence analysis indicated that the aminoacid sequences of *SgHKT1;2* from *S. galapagense* (previously included in the species *S. cheesmaniae*) and *S. lycopersicum* were identical, while *SgHKT1;1* has a substitution in the amino acid sequence (V/L) in the transmembrane M1_B_ helix region as compared with *SlHKT1;1* [17]. The open reading frames of tomato *HKT1;2* and *HKT1;1* only share 47% nucleotides. Among the differences at the protein level, one of them involves the amino acid before the conserved CGF sequence in the P_B_ pore-loop-domain, that is, an asparagine in the former versus a serine (S) in the latter, which could be functionally important. In class I HKTs, the ability to transport Na^+^ or K^+^, apart from the SGGG configuration in the four pores, depends on the presence of an Aspartic (D) in that position of P_B_, as occurs in the halophyte variant EsHKT1,2, which transports K^+^, while in glycophytes, there is an Asn (N), typical from canonical class I Na^+^-selective transporter [25]. However, the functional implication of a serine at that position in HKT1;1 has certainly not been explored yet. Additionally, the promoter regions of *Sl/SgHKT1;1* revealed more polymorphisms, affecting more predicted *cis*-elements than those of *SgHKT1;2/SlHKT1;2*, especially TCA-element (salicylic acid responsiveness), SORLREP3AT (overrepresented in light repressed promoters), and PREATPRODH (Pro- or hypoosmolarity-responsive element) regulatory elements. These differences in the promoter region probably explain their complex pattern of tissular expression. In previous studies, *SgHKT1;2* expression was much lower in roots and higher in leaves, unlike *SlHKT1;2*, whose expression follows an opposite pattern [17,23,24]. Another difference is its apparent lack of change in transcription level at the root due to salinity (70 mM NaCl, in [24]) or even a decrease in leaves where it is expressed most [23]. Regarding *HKT1;1*, its expression is very low for both alleles in comparison to *HKT1;2* [17], higher in root than leaf, and higher for the galapagense allele than for the lycopersicum allele. Differences in the letter sizes of their names in Figure 5 try to represent these differential tissular expressions.

### 3.1. A Model for Na Recirculation

Our statistical approach to study the silencing effects separately (Figure 4 and Appendix A) allows the interpretation of the SgHKT1;1 and SgHKT1;2 roles regarding the distribution of the Na content between mature leaves and fruits under salinity (70 mM NaCl). This interpretation has been schematically represented in Figure 5. Silencing *SgHKT1;2* at the root (black arrows in R2 × S1 × E and R2 × S2 × E interactions for L_Na and F_Na) increases both L_Na (Figure 4C,E, respectively) and F_Na (Figure 4D,F, respectively) suggesting that the xylem Na unloading at the root is the main function here. The significant interaction of R1 × R2 × E for L_Na (Figure 4H), but not for F_Na, also supports the hypothesis of xylem Na unloading for HKT1;2 and suggests xylem Na loading for HKT1;1 at the root under salinity. Silencing *SgHKT1;2* at the scion (green arrows in S1 × S2 × E for L_Na and F_Na in Figure 4A,B, respectively) increases L_Na while decreasing F_Na when SgHKT1;1 is also silenced at the scion, suggesting phloem Na loading in the scion. A similar interpretation derives from silencing SgHKT1;2 at the scion in R2 × S2 × E interaction for L_Na (increasing effect in Figure 4E) and F_Na (decreasing effect when SgHKT1;2 is also silenced at the root in Figure 4F). Silencing *SgHKT1;1* at the scion (orange arrows in S1 × S2 × E interactions for L_Na and F_Na in Figure 4A,B, respectively) decreases L_Na and increases F_Na, suggesting phloem Na unloading in the scion. Silencing *SgHKT1;1* at the root (blue arrows in R1 × S1 and R1 × R2 × E interactions for L_Na in Figure 3D and Figure 4H, respectively) also decreases L_Na, suggesting the xylem Na loading function for HKT1;1 at the root.

Therefore, regarding Na distribution under salinity, the directions of silencing effects of *SgHKT1;1* (orange arrows) and *SgHKT1;2* (green arrows) at the scion are opposite and depend on the given organ (source-leaf, L, or sink-fruit, F, in Figure 4) suggesting phloem Na unload for SgHKT1;1, and phloem Na load for SgHKT1;2 as their relevant roles in leaves, where most nutrient transfer from xylem to phloem occurs (leaf circle in Figure 5). However, in the case of the rootstock, silencing *SgHKT1;2* increases the Na content in all scion tissues (black arrows), indicating its major role is Na xylem unload, while the silencing *SgHKT1;1* decreases L_Na (blue arrows in Figure 4H and Appendix A), suggesting xylem Na load; i.e., opposite directions in the flux of Na in the root, where most nutrient transfer from phloem to xylem occurs (root circle in Figure 5). Following this model, a consequent accumulation of Na in the cortical cells of the root should be expected under salinity when the rootstock is silenced at HKT1;1. In this situation, it is conceivable that Na concentrates in the vacuole reaching concentrations that could favour the entry of water by osmosis and limit the additional intake of sodium for passive transport through a concentration gradient, which would explain the beneficious effects of using the silenced *HKT1;1* line (h1) as rootstock on the vegetative growth of both parts of the plant (APDW and RDW in Appendix A) and, consequently, in fruit yield. We did not measure the root Na contents in the present experiment, but we measured it in a previous experiment using the HKT1;2 silenced lines [24], and a reduction in their root Na content was observed, as expected if the model were true. The role of SgHKT1;2 in Na distribution coincides with that of our previous model proposed for ScHKT1;2, in which this transporter is involved in xylem Na unloading and phloem Na loading [29]. However, the models disagree regarding the role of HKT1;1 in Na distribution. In the present model, SgHKT1;1 would appear to operate in reverse for xylem Na loading and phloem Na unloading (Figure 5), at least under salinity conditions, which would move the cation in the opposite direction of how HKT transporters function, placing the cation inside the cell. Our present results regarding *SgHKT1;1* recall those obtained by [27], who showed that RNAi-mediated down-regulation of *HvHKT1;5* from barley also results in salt tolerance. Thus, HvHKT1;5 performed different functions in barley as compared to its homologous genes in rice and wheat. Ref. [27] suggested that the different localizations of HvHKT1;5 around the xylem mainly leads Na transport to occur in the opposite direction and to a different physiological function in barley. Tomato HKT1;1 and HKT1;2 perform the same Na transporting function [17,20], while both *HKT1;1* and *HKT1;2* gene expressions could be detected in the cells of vascular bundles of the main and secondary veins of tomato leaf; however, only HKT1;2 could be detected in the stellar cells of root tissues, unlike HKT1;1, whose expression was undetectable in roots using in situ PCR [23]. HKT1;1 could be localized in cells adjacent to the xylem and phloem vessels, which differs from HKT1;2 localizations. As with HvHKT1;5, these distinct cellular localizations could explain why the function of SgHKT1;1 in Na translocation from roots to shoots in tomato differs from tomato HKT1;2 and other plant HKT1-like transporters.

### 3.2. Leaf K Content Might Regulate Fruit Soluble Solids Content under Salinity

There is a controversy regarding the use of mild abiotic stresses (drought and salinity) to increase the quality of tomato fruits because these quality increments could be explained by reductions in fresh fruit weight. In addition, marker-assisted selection based on the knowledge of tomato SSC QTLs appears to be inefficient for cultural areas that are variable in salinity level [31]. A new approach for increasing fruit quality based on present results could be useful. Thus, the silenced *SgHKT1;1* at the scion seems to be beneficial for fruit quality (SSC and β-carotenoids, in Figure 3, Appendix A) but only when *SgHKT1;2* remains wild type, suggesting that the described benefits on fruit quality may not come just by grafting a commercial variety on a silenced *SgHKT1;1* rootstock, although it is something to be tested. Our results regarding the relationship between HKT1 transporters and SSC and β-carotenoid accumulation in the fruit under salinity are intriguing. Differences in K distribution might be involved, at least, in SSC changes. Potassium and photoassimilates are loaded together in source tissues and downloaded into sinks [32]. Membrane depolarization caused by sucrose transport is prevented by the release of K^+^ by the plasma membrane outward K^+^ channel AKT2 [32]. Thus, L_K reduction when both scion *SgHKT1* genes are silenced (S1 × S2 interaction for L_K in Appendix A) is accompanied by an SSC reduction in the fruit (S1 × S2 interaction for SSC in Appendix A), suggesting that the level of leaf K content could, positively or negatively, regulate the fruit soluble solids content of the fruit. Multiple reports suggest K^+^ could play an important role in cell signalling when under salinity [33].

The K contents of the tested tissues and xylem sap are clearly affected by the silencing of *SgHKT1* genes (Appendix A), even under control conditions (see triple interactions S1 × S2 × E, R1 × S1 × E and R1 × R2 × E in Appendix A) where the presence of effects of silencing the gene at one locus depends on the genotype at the other, even at different organs (R1 × S1 × E interaction for L_K). Under salinity, the directions of silencing effects are maintained, but the effects of silencing *SgHKT1;1* at the scion (orange arrows) are bidirectional (see triple interactions S1 × S2 × E for X_K and F_K, and R1 × S1 × E for L_K in Appendix A). In the case of the S1 × S2 × E interaction, the same bidirectional changes are observed for X_K and F_K. This could be the reason for the strong correlation found between X_K and F_K (r = 0.87) only under salinity. Here, [K^+^] in the xylem sap increases when both scion *SgHKT1* genes are silenced but decreases when *SgHKT1;2* is Wt. Similarly, when silencing *SgHKT1;1* at the scion but not in the rootstock, L_K decreases, while L_K increases when root *SgHKT1;1* is silenced (interaction R1 × S1 × E for L_K in Appendix A) under salinity. Therefore, *SgHKT1;1* seems to be, at least indirectly, involved in the regulation of K nutrition under salinity. In fact, the beneficial effect of silencing *SgHKT1;1* at the root is also clear from the K perspective. Thus, h1/h1 and h1/Wt rootstock/scion combinations are the best maintaining L_K under salinity (Appendix A), and parallel changes between L_K and RDW and TFW were observed (blue arrows in R2 × R1 interactions for L_K and RDW, and interactions S1 × R1 for L_ K and TFW in Appendix A) in agreement with the grouping of L_K, RDW and TFW after the Principal Component Analysis of the salinity data (Figure 2). Apparently, K mitigates the detrimental effects of salinity, such as reduced water availability, Na^+^ toxicity to cell metabolism, and increased ROS production [34]. Now we add additional evidence showing that the effects of silencing Na transporters encoding genes *SgHKT1;2* and *SgHKT1;1*, particularly under salinity (Na excess), not only affects Na fluxes (as indicated in our model in Figure 5) but also K fluxes, generally following an opposite direction [23,24,29]. Potassium is highly mobile within the plant, exhibiting cycling between roots and shoots via xylem and phloem [35], where potassium channels SKOR and AKT2 play an important role, particularly under salinity, since changes in [K^+^], ABA, and ROS affect SKOR xylem activity/expression, and ABA influences AKT2 phloem expression [32]. NIL14 carries the low producing ABA *S. galapagense* allele at gABA1, a QTL in chromosome 1 detected for ABA concentration in the xylem sap when using a RIL population derived from *S. galapagense* as rootstock for QTL analysis of salt tolerance [16], so changes in [K^+^], and ROS could be the major players in our reciprocal grafting experiment with *SgHKT1* silenced lines under salinity.

Plants employ antioxidant compounds such as phenolic compounds, ascorbic acid, tocopherols, glutathione, and carotenoids to eliminate reactive oxygen species [36]. As a side effect, their accumulation in the fruit can increase its nutraceutical value. For this reason, Ref. [37] suggested using salt stress to enhance fruit quality. Noteworthy, we have observed that the changes affecting the fruit content of the antioxidant compound β-carotenoids (and not affected by salinity) depended on the silencing genotype at HKT1;2 (downwards green arrows in Appendix A), and mainly, at HKT1;1 (blue and orange upwards arrows in Figure 3 and Appendix A). Therefore, similarly to the fruit content of SSC, silencing *SgHKT1;1* has an increasing effect on fruit β-carotenoids.

In conclusion, we have shown that SgHKT1;1 is functionally different from SgHKT1;2 regarding Na circulation in the tomato vascular system allowing us to build a model that explains that using the silenced *SgHKT1;1* line as rootstock would improve salt tolerance and fruit quality of varieties carrying the wild type *SgHKT1;2* allele. Moreover, the increasing effect on yield and SSC of silencing *SgHKT1;1* could explain that a low expressing *HKT1;1* variant in *S. lycopersicum* was fixed during the domestication, and the paradox of increasing salt tolerance through silencing the *HKT1;1* allele from *S. galapagense*, a salt adapted species.

## 4. Materials and Methods

### 4.1. Plant Materials

In order to have all four combinations (for silenced and non-silenced genes at HKT1;1 and HKT1;2), three homozygous transgenic lines were obtained from the near-isogenic line NIL14 as described in [29]: line h1, silenced at HKT1;1, line h2, silenced at HKT1;2, and line h1-h2, silenced at both HKT1 genes located in tandem in chromosome 7. The fourth combination, NIL14 (Wt in this study; i.e., with non-silenced genes at both loci), was obtained after 5 selfing generations of an F_1_ progeny from a cross between a salt-sensitive genotype of *S. lycopersicum*, var. Cerasiform (E9) as the female parent, and the salt-tolerant accession L2 [4] from *S. cheesmaniae* (L. Riley) Fosberg as a male parent [17]. This L2 line has been now re-classified as *S. galapagense* following the taxonomic study by [38]. Seed viability of L2 had been lost, but after using tissue culture technology, this line recovered its propagation capacity by seed. Thus, we could observe that plants from L2 showed the typical leaf morphology of *S. galapagense*, producing 3 orders of leaflets [38,39] (Appendix A). NIL14 and parents were genotyped for the 7720 SolCAP panel of SNPs. Both parents (E9 and L2) share genotype at 4800 SNP loci. NIL14 presents E9 genotype at 1919 loci and *S. galapagense* genotype at 981 loci, including HKT1;1 and HKT1;2 loci. Regarding other genes involved in Na^+^ homeostasis, this NIL is homozygous for the *S. galapagense* allele at SOS1 and NHX4 and homozygous for the E9 allele at HAK20 [40] as inferred from the genotypes at the flanking SNPs (solcap_snp_sl_64082 and solcap_snp_sl_21335) of Soly04g008450.2. All 16 rootstock/scion graft combinations involving non-silenced (Wt), single RNAi-silenced lines for *SgHKT1;1* (h1), *SgHKT1;2* (h2), as well as the doubly silenced line at both loci (h1-h2) were used for data analysis. The same nomenclature for plants (as rootstock/scion combinations of HKT1;1 and HKT1;2 genotypes) as in [29] was used here. Then, R1 and R2 denote the effect of gene silencing (vs. non-silencing) at SgHKT1;1 and SgHKT1;2 loci, respectively, in the rootstock; similarly, for S1 and S2 in the scion.

### 4.2. Plant Growth Conditions and Phenotyping

Methods related to the grafting and culture of plants were extensively described in [29]. All genotype-rootstock-scion combinations and their reciprocals (16) were cultured under two soil environmental treatments (E), control and salinity (0 or 70 mM NaCl) supplied to the irrigation solution. After 60 days of salt treatment, the xylem sap of six plants per graft combination and environmental treatment was collected and stored at 4 °C for further analysis. Then, the aerial parts (stems, leaves, and flowers) and washed roots were dried at 65 °C for 72 h in a forced-air oven, followed by measurement of dry mass. The dry biomass of the aerial part (APDW) was calculated as the sum of the dry biomasses of leaves + stems + flowers and expressed as g/plant. The aerial part to root (AP/R) ratio was calculated by dividing aerial part dry weight (APDW) by root dry weight (RDW). Na^+^ and K^+^ contents in the xylem sap (expressed as µg/mL), flowers in the 3rd inflorescence (including peduncles, pedicels, and flowers) and in the leaf directly below this inflorescence (9th leaf), and fruits (expressed as g/100 g of the tissular dry weight) were analysed for ion composition using a Varian 720-E inductively coupled plasma-optical emission spectrometer (ICP-OES; Scientific Instrumentation Service, EEZ, CSIC, Granada, Spain).

Fruits from the first three trusses were harvested while maturing from the end of February 2019 up to the beginning of April 2019 (after between 105 and 145 days from the beginning of the salt treatment). The total number of fruits (TFN), expressed as the number of fruits/plant, and total fruit weight (TFW), expressed as g/plant, were determined for six plants per genotype, graft combination, and salt treatment. The soluble sugar content (SSC -°Brix), titrable acidity (A, %), β-carotenoid (BCarot., mg/100 g of the fresh weight), and total phenolic compounds (phenolics, mg GAE/100 g of the fresh weight) were determined as described by [24].

### 4.3. Statistical Analysis

To study the association among the different traits, Pearson correlation and Principal Component analyses were used within each environmental treatment (control and salinity).

A five-way 2^5^ full factorial fixed effects linear model was used for the data analysis. Main effects and interactions were estimated to test for the effects of silencing (vs. non-silencing) the genes *SgHKT1;1* and *SgHKT1;2* in all rootstock (R1 and R2, respectively) and scion (S1 and S2, respectively) combinations, and the saline treatment (E) on evaluated traits. Statistical significance was considered at the conventional 5% level (*p* ≤ 0.05). A significant interaction will imply that the effect of silencing in one factor depends on the levels of the other factor. For instance, a significant R1 × R2 interaction means that the effect of silencing the gene *SgHKT1;1* in the rootstock will depend on the genotype at the other locus *SgHKT1;2* in the rootstock; similarly, a significant R1 × E indicates that the effect of silencing the gene *SgHKT1;1* in the rootstock will depend on the saline treatment. For significant interactions, mean comparisons by LSD Fisher (α = 0.05) were graphed using adjusted means and standard errors to observe the direction of silencing effects. The Infostat statistical package version 2020 (Centro de Transferencia InfoStat, FCA, Universidad Nacional de Córdoba, Argentina) [41] was used for data analysis and graphs.

## Figures and Tables

**Figure 1 ijms-23-05130-f001:**
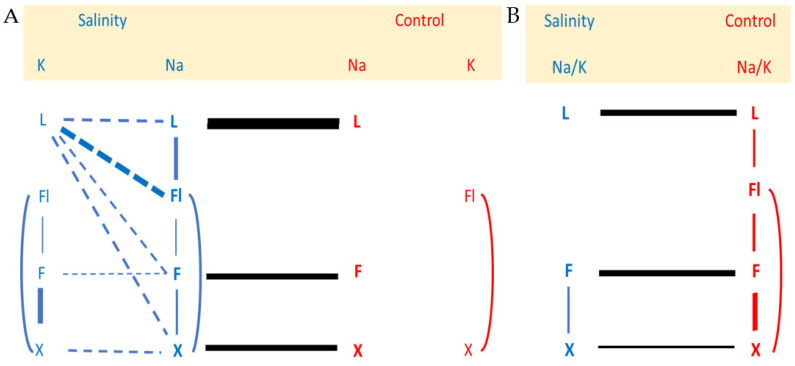
Schematic representation of significant (*p* < 0.05) correlations by a continuous line (positive correlation), or a discontinuous line (negative correlation) connecting Na and K contents (**A**), and The Na/K ratio (**B**), of different organs (L: leaf, Fl: inflorescence, F: fruit, X: xylem sap). The thicker the line, the higher the correlation coefficient (see Appendix A).

**Figure 2 ijms-23-05130-f002:**
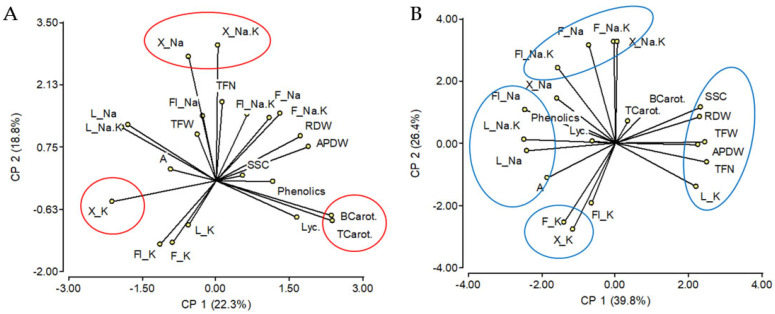
Graphic representation (biplot) of principal component analysis of variability found among the 16 rootstock/scion SgHKT1 genotypic combinations under (**A**) control condition (0 mM NaCl), and (**B**) salinity (70 mM NaCl) for evaluated traits (agronomical, physiological, and vegetative plant traits). Closely related traits are encircled.

**Figure 3 ijms-23-05130-f003:**
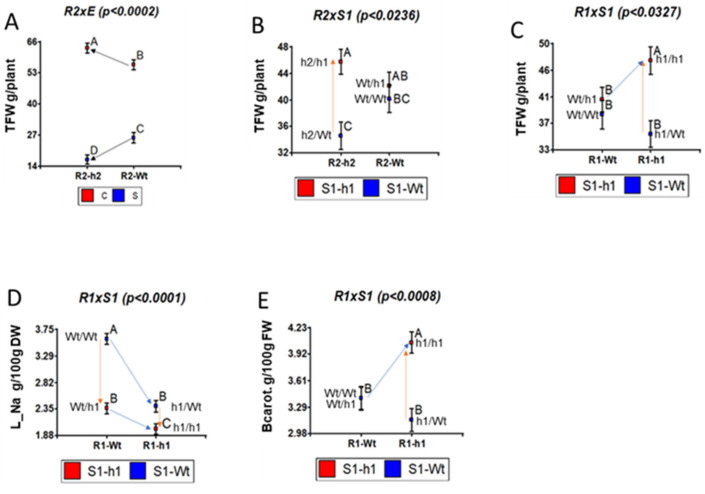
Graphic representation of relevant interactions involving rootstock genotype at HKT1.1 (R1) and HKT1.2 (R2) and salinity treatment (E) for total fruit yield (TFW): (**A**) R2 × E, (**B**) R2 × S1, and (**C**) R1 × S1. (**D**) R1 × S1 interaction for leaf Na content (L_Na in g/100 g of dry weight), and (**E**) fruit β-carotenoids (BCarot., mg/100 g fresh weight). Means with different letters are significantly different at 5% level. Arrows show the directions of the effects of silencing *SgHKT1;1* at the scion (orange arrows) and at the root (blue arrows), and silencing *SgHKT1;2* at the root (black arrows).

**Figure 4 ijms-23-05130-f004:**
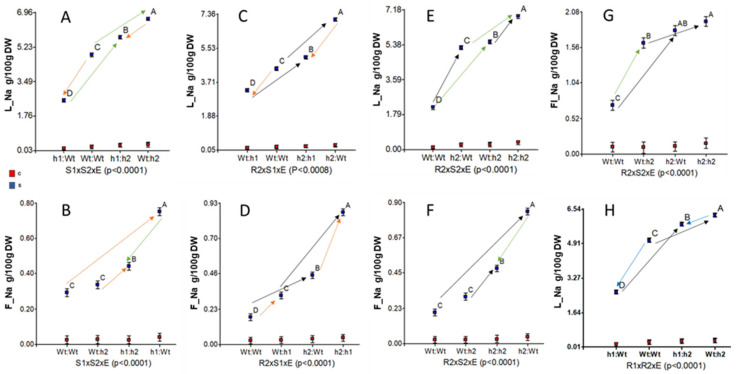
Graphical representation of triple interactions involving the salinity level for the Na content of leaf (L_Na, in **A**,**C**,**E**,**H**), fruit (F_Na, in **B**,**D**,**F**) and inflorescence (Fl_Na, in **G**). S1 × S2 × E interactions in (**A**,**B**). R2 × S1 × E interactions in (**C**,**D**). R2 × S2 × E interactions in (**E**, **F** and **G**). R1 × R2 × E interaction for L_Na in (**H**). In all cases, no mean differences were found for control conditions (red squares). Arrows show the directions of the effects of silencing *SgHKT1;1* at the scion (orange arrows) and at the root (blue arrows), and silencing *SgHKT1;2* at the scion (green arrows) and at the root (black arrows). Means with different letters are significantly different at 5% level.

**Figure 5 ijms-23-05130-f005:**
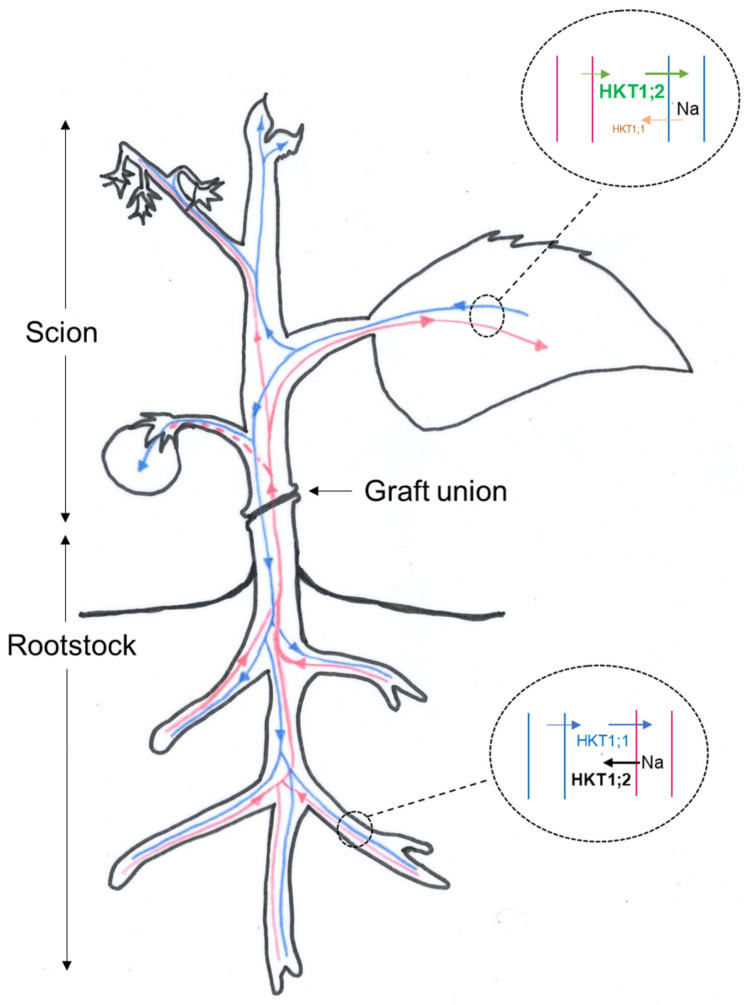
A model for Na recirculation under salinity based on the results of this reciprocal grafting experiment, the previous knowledge of HKT1 transporters, and the main features of plant vasculature. The diagram shows the directions of xylem (red) and phloem (blue) transport of nutrients in a grafted plant. A discontinues red line to the fruit indicates that the xylem sap [K^+^] and the fruit K content are highly correlated under salinity. The direction of Na flux in the vasculature of the mature leaf (where most xylem-to-phloem transfer occurs), and that of the root (where most phloem-to-xylem transfer takes place) are schematically represented within the respective circles, as inferred from the silenced effects of *SgHKT1;1* at the scion (orange arrows) and at the root (blue arrows), and silencing *SgHKT1;2* at the scion (green arrows) and at the root (black arrows). Letter size of transporters is indicative of their differential expression in leaf and root [17,23].

## Data Availability

Not applicable.

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
