# Peer review of "HKT1;1 and HKT1;2 Na+ Transporters from Solanum galapagense Play Different Roles in the Plant Na+ Distribution under Salinity"

_ijms, 2022, doi:10.3390/ijms23095130_

Round 1
Reviewer 1 Report
This manuscript described how HKT1;1 and HKT1;2 Na+ transporters from Solanum gala-pagense play different roles in the plant Na+ distribution under
salinity.
Although the authors investigated many molecular data their role in altering plant physiology is missing. Some of the parameters like osmolytes regulation, electrolyte leakage, and antioxidant defense would be effective in drawing a complete conclusion.
The discussion should be improved by linking the parameters.
Some of the references are old. Please cite recent references on salt stress responses and tolerance.
Author Response
Response to Comments and Suggestions for Authors by Reviewer 1
This manuscript described how HKT1;1 and HKT1;2 Na+ transporters from Solanum gala-pagense play different roles in the plant Na+ distribution under
salinity.
Although the authors investigated many molecular data their role in altering plant physiology is missing. Some of the parameters like osmolytes regulation, electrolyte leakage, and antioxidant defense would be effective in drawing a complete conclusion.
The discussion should be improved by linking the parameters.
It is well known that salt stress results in osmotic stress, nutrient imbalances, oxidative damage, reduced ROS scavenging capacity, and decreased stomatal aperture and photosynthetic activity, and that salt tolerance is achieved by regulating Na+ and K+ transport, accumulating different osmoprotectants and reducing ROS toxicity (i.e., Devireddy, A. R., Zandalinas, S. I., Fichman, Y., & Mittler, R. (2021). Integration of reactive oxygen species and hormone signaling during abiotic stress. The Plant Journal, 105, 459-476). We completely agree that measuring those parameters would have enriched the discussion and conclusions, however, it would been unmanageable for us because the size and complexity of our experiment (6x16x2=192 grafted plants under agronomic and physiological evaluation). In this work, we have evaluated the roles of HKT1 transporters in planta regarding the dynamics of Na and K fluxes in xylem and phloem and the effects of their silencing on important recognized physiological parameters of salt tolerance such as Na/K homeostasis, vegetative growth of the aerial part and production of fruits in a commercial greenhouse, under real production conditions.
Some of the references are old. Please cite recent references on salt stress responses and tolerance.
The older references contain original results whose specific conclusions support our own and are suitable for the discussion. The rest of the citations are mostly quite recent, between 2018-2021. Among cited references on salt stress response and tolerance are [6] (2022), [19] (2020), [20] (2021) and [33] (2019).
We would like to thank you for your assistance in improving the manuscript and hope that the revised manuscript is now acceptable for publication in IJMS
Yours sincerely,
Dr. Maria J. Asins

Reviewer 2 Report
In this work the authors address the study of salt tolerance using isogenic lines obtained for Solanum galapagense alleles encoding both class I Na+ transporters HKT1;1 and HKT1;2 to investigate the silencing effects on the Na and K contents of the xylem sap, and source and sink organs of the scion, and their contribution to salt tolerance in all rootstock/scion combinations of non-silenced and silenced lines, under two salinity treatments. The experimentation was well performed and the methods adopted are reliable and effective, to provide new insight on the molecular mechanism and transcriptional regulation about HKT1;1 and HKT1;2 genes encode class I HKT Na+-selective transporters which are expressed in the vascular system, in the xylem and likely in the phloem of tomato root and shoot tissues.
The work is original and can be accepted if the authors respond to the following reviewer's comments.
The authors make 16 rootstock/scion combinations of silenced and wild type S. galapagense HKT1 lines but data analysis and interpretation becomes particularly complex.
An interesting aspect is: in the main component analysis galapagense lines HKT1, SSC was highly and positively related to fruit yield and vegetative growth only under salinity. Therefore, a better vegetative growth was linked not only to a higher yield of fruit, but also to a greater SSC of fruits in these materials under salinity. Can the authors suggest a physiological explanation for this experimental response?
The results obtained induce the authors to represent a model of vascular transfer of sodium flow:
in the mature leaf occurs most of the xylem-phloem transfer while in the root most of the transfer from phloem to xylem as deduced from the silenced effects of SgHKT1;1 at the scion and root.
Such an interesting model would suggest an accumulation of Na in the cortical cells of the root. Since the authors do not dose sodium in the root, it is conceivable that it concentrates in the vacuole reaching concentrations such as to favor the entry of water by osmosis and to limit the additional intake of sodium for passive transport through a concentration gradient.
Another interesting aspect is that the tissular distribution of K is directly, or indirectly, regulated by genotypic interactions between SgHKT1;1 and SgHKT1;2.
How do the authors explain that the content of K in the leaf is not related to the content of K in other tissues?
Because silencing S. galapagense alleles at HKT1;1 and HKT1;2 loci affected the leaf K content Is it possible to conceivable an influence also on the stomatal regulation?
Finally, I ask the authors whether the theoretical model that explains that using the silenced SgHKT1;1 line as rootstock that would improve salt tolerance and fruit quality of varieties carrying wild type SgHKT1;2 allele could in the future work have a verification in the field ?
Author Response
Response to Comments and Suggestions for Authors by Reviewer 2
In this work the authors address the study of salt tolerance using isogenic lines obtained for Solanum galapagense alleles encoding both class I Na+ transporters HKT1;1 and HKT1;2 to investigate the silencing effects on the Na and K contents of the xylem sap, and source and sink organs of the scion, and their contribution to salt tolerance in all rootstock/scion combinations of non-silenced and silenced lines, under two salinity treatments. The experimentation was well performed and the methods adopted are reliable and effective, to provide new insight on the molecular mechanism and transcriptional regulation about HKT1;1 and HKT1;2 genes encode class I HKT Na+-selective transporters which are expressed in the vascular system, in the xylem and likely in the phloem of tomato root and shoot tissues.
The work is original and can be accepted if the authors respond to the following reviewer's comments.
The authors make 16 rootstock/scion combinations of silenced and wild type S. galapagense HKT1 lines but data analysis and interpretation becomes particularly complex.
An interesting aspect is: in the main component analysis galapagense lines HKT1, SSC was highly and positively related to fruit yield and vegetative growth only under salinity. Therefore, a better vegetative growth was linked not only to a higher yield of fruit, but also to a greater SSC of fruits in these materials under salinity. Can the authors suggest a physiological explanation for this experimental response?
The origin of such association could come from the particular set of materials used in this salt tolerance experiment: all possible rootstock/scion graft combinations involving non silenced, single silenced and doubly silenced line for both SgHKT1;1 and SgHKT1;2 loci. Since these targeted loci are the major factors (likely the only ones here) controlling Na/K homeostasis in these tomato plants, and this is a key parameter for salt tolerance, the positive association of fruit yield (and vegetative growth) with SSC of fruits under salinity seems reasonable through the preservation of the photosynthetic machinery (maintaining leaf K content high), allowing photosynthesis and carbohydrate production in the leaves. Certainly, leaf K content is also associated with both SSC of the fruit and yield under salinity.
The results obtained induce the authors to represent a model of vascular transfer of sodium flow:
in the mature leaf occurs most of the xylem-phloem transfer while in the root most of the transfer from phloem to xylem as deduced from the silenced effects of SgHKT1;1 at the scion and root.
Such an interesting model would suggest an accumulation of Na in the cortical cells of the root. Since the authors do not dose sodium in the root, it is conceivable that it concentrates in the vacuole reaching concentrations such as to favor the entry of water by osmosis and to limit the additional intake of sodium for passive transport through a concentration gradient.
Absolutely. Following the model, the Na content of the root in the HKT1;1 silenced rootstock should increase under salinity, and the double effect you mention, favoring the entry of water by osmosis and limiting the additional intake of sodium for passive transport through a concentration gradient, would better explain the beneficious effects of using silenced HKT1;1 rootstock on vegetative growth of both parts of the plant (particularly the root in h1/Wt) and, consequently, in fruit yield (we have added this relevant idea to the discussion, lines 376-386). We did not measure root Na contents in present experiment but, in a previous one, using the HKT1;2 silenced lines (Romero-Aranda et al. 2020) we did it, and a reduction of the root Na content was observed in the silenced HKT1;2 lines, as expected if the model were true.
Another interesting aspect is that the tissular distribution of K is directly, or indirectly, regulated by genotypic interactions between SgHKT1;1 and SgHKT1;2.
How do the authors explain that the content of K in the leaf is not related to the content of K in other tissues?
Potassium is an extremely mobile element, essential for the physiology of the plant which has evolved sophisticated mechanisms for the uptake and distribution of K+ both within cells and between organs [35]. K is supplied to the leaf through the xylem (input 1) and is exported from the leaf through the phloem (output), a fraction of which may come back to the leaf again through the xylem (input 2). Therefore, leaf K content depends on at least three variables. The lack of significant correlation between leaf K content and xylem sap [K], for example, could be due to counteracting variation between inputs and output components when considering the whole set of rootstock/scion combinations.
Because silencing S. galapagense alleles at HKT1;1 and HKT1;2 loci affected the leaf K content Is it possible to conceivable an influence also on the stomatal regulation?
Yes, K play an important role in turgor regulation within the guard cells during stomatal movement, since silencing SgHKT1 genes affects leaf K content, even under control conditions, it would be possible to conceivable an influence also on the stomatal regulation. We did not measure stomatal conductance in present experiment, nevertheless, in a previous one using HKT1;2 silenced lines (Romero-Aranda et al. 2020), no significant change for stomatal conductance after 34 days of salt treatment was observed.
Finally, I ask the authors whether the theoretical model that explains that using the silenced SgHKT1;1 line as rootstock that would improve salt tolerance and fruit quality of varieties carrying wild type SgHKT1;2 allele could in the future work have a verification in the field ?
Yes, that is the main purpose of the model, to verify in the field that using a silenced HKT1;1 line as rootstock, the salt tolerance and fruit quality of the grafted variety would improve. First, we are going to compare the conclusions of this experiment built up on SgHKT1 alleles to those from another one based on S. lycopersicum alleles at HKT1 loci. If the conclusions are the same then, silenced Sg/SlHKT1;1 rootstocks would improve salt tolerance and fruit quality of commercial varieties (carrying lycopersicum alleles at both HKT1 loci) what, anyhow, will be tested in a salt tolerance experiment using the silenced SgHKT1;1 line as rootstock, likely this year. The worst scenario would be that, as predicted by our model, the grafted variety should have to express SgHKT1;2 in their leaves. The galapagense and lycopersicum alleles at HKT1;2 have the same sequences at their open reading frames but differ in their promoter sequences. This difference at the promoter determines their differential patterns of tissue expression. Since introgression of SgHKT1;2 into elite varieties without introgressing SgHKT1;1 is not possible due to their extreme linkage, the promoter of SlHKT1;2 in the variety should have to be changed to increase leaf (scion) expression of SlHKT1;2 by prime editing techniques. The resulting tomato variety, highly expressing SlHKT1;2 in the leaves, grafted on a silenced HKT1;1 rootstock should work in the field, if not, the model is wrong.
We would like to thank you for your assistance in improving the manuscript and hope that the revised manuscript is now acceptable for publication in IJMS
Yours sincerely,
Dr. Maria J. Asins

Reviewer 3 Report
The manuscript shows information on the tolerance to salt stress by wild varieties of tomato. The authors mainly analyzed the role of Na transporters and their contribution to tolerance to salt stress. In general, the manuscript is framed in the problem of tolerance to environmental stress, mainly salt stress, one of the most important issues today. The authors focused mainly on two genes (HKT1;1 and HKT1;2) that regulate the homeostasis of Na and K in the shoot. Based on previous work, they have deepened the role of the two transporters and the possibility that they can contribute to the protection of plants in environments with excess salt.
The manuscript is well written and clear. The analyses were performed appropriately and most importantly the manuscript attempts a significant correlative approach to understand the role that different alleles can have in tolerance to salt stress. Instead of focusing exclusively on molecular analysis, the authors attempted a correlative approach by measuring the relative input of mutant lines and using a system based on rootstocks and scions. I really like this kind of approach that immediately attempts a correlation between genes and phenotype.
Sincerely, I am very much in favor of the approval and publication of the manuscript in its current form. While rereading it, I do not find any criticism, if not minimal and of little importance. At the end of the manuscript, the authors propose a model that correctly interprets the available data and proposes the use of particular lines for future breeding in order to increase tolerance to salt stress in tomatoes.
Author Response
Response to Comments and Suggestions for Authors by Reviewer 3
The manuscript shows information on the tolerance to salt stress by wild varieties of tomato. The authors mainly analyzed the role of Na transporters and their contribution to tolerance to salt stress. In general, the manuscript is framed in the problem of tolerance to environmental stress, mainly salt stress, one of the most important issues today. The authors focused mainly on two genes (HKT1;1 and HKT1;2) that regulate the homeostasis of Na and K in the shoot. Based on previous work, they have deepened the role of the two transporters and the possibility that they can contribute to the protection of plants in environments with excess salt.
The manuscript is well written and clear. The analyses were performed appropriately and most importantly the manuscript attempts a significant correlative approach to understand the role that different alleles can have in tolerance to salt stress. Instead of focusing exclusively on molecular analysis, the authors attempted a correlative approach by measuring the relative input of mutant lines and using a system based on rootstocks and scions. I really like this kind of approach that immediately attempts a correlation between genes and phenotype.
Sincerely, I am very much in favor of the approval and publication of the manuscript in its current form. While rereading it, I do not find any criticism, if not minimal and of little importance. At the end of the manuscript, the authors propose a model that correctly interprets the available data and proposes the use of particular lines for future breeding in order to increase tolerance to salt stress in tomatoes.
We deeply appreciate your comments and hope that the revised manuscript is now acceptable for publication in IJMS
Yours sincerely,
Dr. Maria J. Asins
